# A Novel Representation of Audiological and Subjective Findings for Acoustical, Bone Conduction and Direct Drive Hearing Solutions

**DOI:** 10.3390/jpm13030462

**Published:** 2023-03-01

**Authors:** Georg Mathias Sprinzl, Astrid Magele, Philipp Schoerg, Rudolf Hagen, Kristen Rak, Anja Kurz, Paul Van de Heyning, Miryam Calvino, Luis Lassaletta, Javier Gavilán

**Affiliations:** 1Department of Otorhinolaryngology, Head & Neck Surgery, University Clinic St. Poelten, Dunant-Platz 1, 3100 St. Pölten, Austria; 2Karl Landsteiner Institute of Implantable Hearing Devices, 3100 St. Pölten, Austria; 3Department of Oto-Rhino-Laryngology, Head and Neck Surgery, Comprehensive Hearing Center, University of Wuerzburg, Josef-Schneider-Str. 11, 97080 Würzburg, Germany; 4ENT Department, Antwerp University Hospital, Wilrijkstraat 10, 2650 Antwerp, Belgium; 5Department of Otolaryngology, La Paz University Hospital, 28046 Madrid, Spain

**Keywords:** bone conduction implant, hearing aids

## Abstract

Background: The benefit of hearing rehabilitation is often measured using audiological tests or subjective questionnaires/interviews. It is important to consider both aspects in order to evaluate the overall benefits. Currently, there is no standardized method for reporting combined audiological and patient reported subjective outcome measures in clinical practice. Therefore, this study focuses on showing the patient’s audiological, as well as subjective outcomes in one graph using data from an existing study. Method: The present paper illustrated a graph presenting data on four quadrants with audiological and subjective findings. These quadrants represented speech comprehension in quiet (unaided vs. aided) as WRS% at 65 dB SPL, speech recognition in noise (unaided vs. aided) as SRT dB SNR, sound field threshold (unaided vs. aided) as PTA_4_ in dB HL, wearing time and patient satisfaction questionnaire results. Results: As an example, the HEARRING graph in this paper represented audiological and subjective datasets on a single patient level or a cohort of patients for an active bone conduction hearing implant solution. The graph offered the option to follow the user’s performance in time. Conclusion: The HEARRING graph allowed representation of a combination of audiological measures with patient reported outcomes in one single graph, indicating the overall benefit of the intervention. In addition, the correlation and consistency between some results (e.g., aided threshold and aided WRS) can be better visualized. Those users who lacked performance benefits on one or more parameters and called for further insight could be visually identified.

## 1. Introduction

Audiological measures used in clinical practice may not reflect a patient’s overall hearing device benefit. Therefore, it is important to consider the patient’s reported subjective outcomes along with audiological benefit. Even if both entities are collected within a cohort simultaneously, outcomes are not presented as synergetic measures. Multiple graphical representations are more difficult to read and do not provide a comprehensive interpretation of the outcomes. Although the HEARRING graph is not intended to replace any existing means of representation of clinical data, it is aimed at providing an additional comprehensive overview of multiple test findings.

Maier et al. [1], introduced minimal reporting standards for middle ear hearing implants in 2018, suggesting standards using the most relevant audiological outcomes. These standards have been used to present the audiological performance in the graph.

To assess subjective benefit, several questionnaires are available, such as AQoL (Assessment of Quality of Life), SSQ (Speech and Spatial Qualities of Hearing), APHAB (Abbreviated Profile of Hearing Aid Benefit), HUI (Health Utilities Index), DOSO (Device Oriented Subjective Outcome Scale), COSI (Client Oriented Scale of Improvement), HAUQ (Hearing Aid User’s Questionnaire), etc. The aim of these assessment questionnaires is to document patient reported outcomes, focusing on required rehabilitation and overall quality of life. 

The Speech, Spatial, and Qualities of Hearing Scale (SSQ) [2,3] is a widely used patient reported questionnaire assessing an individual’s hearing abilities in three domains: (1) speech understanding, (2) spatial domain and (3) the hearing quality.

Among various studies performed recently, wearing time is one factor that has been used very often to measure subjective benefit. 

Several studies [4,5,6,7,8] have been suggesting consistent device use as an important factor determining progress in hearing rehabilitation. At ESPCI, 2019, Teresa Ching presented a survey (which included 37 clinicians) showing the importance of “amount of device use”, which was ranked to be the foremost important factor to evaluate patients’ satisfaction.

The parameter of wearing time was deemed efficient to indicate the users’ listening and wearing comfort. 

The aim of the current study was to merge subjective findings with audiological ones in order to have a better overview about success of treatment at a glance. Therefore, the concept of the HEARRING graph was introduced. We combined audiological and subjective data in one graph and provided an overall picture of patient benefit with hearing device usage. Even though, in the present study, the HEARRING graph did not provide an in-depth quality control of audiological data, it offered a quick visual hint on whether a single patient’s dataset needed to be looked at in more depth. 

## 2. Materials and Methods

The concept of the HEARRING graph was introduced by showing audiological and subjective data extracted from an independent, already published study by the first-author [9]. In this study, the audiological and patient reported outcomes of patients with CHL/MHL, SSD patients implanted with an active BCI were published.

The following findings have been integrated in the graph: 

Sound field thresholds in unaided and aided conditions. The PTA4 is defined as the an average of 0.5, 1, 2, 4 kHz [1].Word recognition scores (WRS) in quiet, in unaided and aided conditions at 65 dB SPL in sound field. Speech Recognition Thresholds (SRT50) in noise in unaided and aided conditions in sound field.Wearing time (hours/day).SSQ12 questionnaire scores.

The HEARRING graph is illustrated as Figure 1: 

The upper right quadrant shows WRS in % at 65 dB SPL in the aided as well as unaided condition in relation to the respective PTA values on the x axis.

The lower right quadrant demonstrates speech performance in noise (in dB SNR) in the unaided and aided conditions. The SRT_50_ (defined as the level needed to understand 50% of words in a list) is tested with a fixed background noise level at 65 dB SPL and varying speech levels (SNR). In the C/MHL group, the noise and speech were presented from the front, whereas in SSD, noise was presented from the normal hearing ear and speech from the deaf ear. In addition, the SNR values are presented in relation to the corresponding PTA values.

On the upper left quadrant, we included the wearing time (hours/day) as obtained from the patient in relation to WRS. On the lower left quadrant, we incorporated the scores (difference between preop and 3 months postop) of the SSQ12 questionnaire. This is because the aim of the treatment option is to maximize speech performance in every situation as well as offering a comfortable, user friendly hearing restoration throughout the day. 

Scores from all subscales, as well as overall score, are presented in the graph to give a clear understanding of patient performance in different situations.

The dotted connection lines between wearing time, WRS or alternatively SRT dB SNR and the PTA4 are intended to graphically identify results performed in the same condition (unaided or aided) by the same patient or by a cohort of users at the same point in time (Figure 1). Snik et al. [10] suggested ideal target aided thresholds between 20 and 35 dB HL, depending on sensorineural loss. This is also a realistic target with respect to the technical possibilities of currently available hearing devices. In addition, referring to the Articulation Index (AI) introduced by Müller and Killion in the 90s, hearing thresholds of about 35dB HL in main audiometric frequencies (i.e., 0.5 to 4 kHz) lead to an AI of about 0.5. This level of AI allows to achieve a WRS of at least 70% or more and a sentence understanding of at least 95% [11,12,13]. These targets have been used by different authors in the literature as ideal minimal standard treatment for hearing device users.

Based on these considerations, the HEARRING group characterized beneficial treatment as follows: speech score in quiet ≥70% [8,9,14,15,16,17,18] and wearing time ≥8 h [19,20,21,22,23,24], which was visualized in the HEARRING graph.

Furthermore, the present paper graphically divides the relation between wearing times and speech in quiet in 5 regions:

1.Region 1 indicated that the users achieved ≥70% word recognition score in quiet and wore the device for ≥8 h/day. This exhibited an ideal situation. The patient reached high listening and acceptable wearing comfort.2.Region 2 represented word recognition score of ≥70% and wearing time of ≤8 h/day. It suggested that an optimal speech comprehension in the speech test was achieved, but the ideal wearing time was not achieved.3.Region 3 identified ≤70% word recognition score and wearing time of ≥8 h/day. This indicated ideal wearing time but a need to achieve better speech scores.4.Region 4 showed ≤70% word recognition score and wearing time between 4 and 8 h/day.5.Region 5 marked ≤70% word recognition score and wearing time between 1 and 4 h/day.

The 4th and 5th regions highlighted users who might need further support. 

There is also a color coding from orange to green along each axis showing the achieved ranges from poor to optimal conditions. 

## 3. Results

The audiological and subjective data for seven subjects were already published using the conventional method of representation. In addition, the HEARRING graph provides a novel overview of the study results. The unaided results could be identified in black dots and the aided ones in red.

The representation of scores from the SSQ12 questionnaire along with the wearing time were shown to give an overview about the patient’s subjective satisfaction with a hearing device.

The lines connected the wearing times (aided), the corresponding speech results in sound field and the corresponding PTA4. 

The audiological datasets measured in the aided condition could be visually compared to the ones measured in the unaided condition. As observed in the graph, the mean unaided PTA4 is 55 dBHL and mean WRS preimplantation is 13.75% at 65 dB SPL. 

Independently of the degree of hearing loss observed in terms of PTA and WRS in quiet in the unaided condition, users of the active bone conduction implant achieved mean speech scores better than 70% at the three month post-operative appointment at the clinic. The mean wearing time of the device was more than 10 h/day (Figure 1; top left).

The graph showed that the treatment option not only effectively restored hearing abilities as early as three months post implantation, but also offered a comfortable intact-skin solution reflected by the users’ high daily wearing time.

An additional aspect could be shown graphically by comparing unaided PTA and unaided SRT values. As is easily observed in the graph, the unaided SRT value is too good for the unaided PTA4 values. This could be due to participation of the non-test ear in SSD subjects, for example. 

## 4. Discussion

As hypothesized, the HEARRING graph was designed to represent results from different audiological and subjective tests in one graph. The graph required no special tools or expertise to be created. Simple test results could be utilized to see patients’ benefits with different treatment options. This allows seamless integration of the graph into the clinics’ patient data analyses. No new studies were conducted to generate data for the present study.

The graph may help in assessing reliability of tests, understanding patient needs, setting realistic goals for patients, changing or modifying treatment, counseling patients, planning effective rehabilitation and motivating patients.

The results with different treatment options could also be observed in a single glance. This might offer the opportunity to create a very simple and clear understanding of comparison of those treatment options and to assess which treatment option provided the best chance of clinical improvement.

The HEARRING graph offers a quick quality control of the data on a single user level. Users’ results which require attention in terms of double-checking different findings can be highlighted. The consistency check of audiological findings was not discussed in this work.

As shown in the publication, the HEARRING graph put PTA and WRS in quiet in relation to wearing times as well as SRT dB SNR performance which reflects the reality better than speech in quiet.

The combination of the two parameters, speech results and wearing time, might also guide the audiologist or hearing aid specialist during the fitting and counseling process. The reasons for low wearing time could be discussed during an interview with the expert, and alternative treatment options might be considered to improve the listening situation in accordance with the user’s needs.

The target is to achieve 70% WRS at 65 dB as this allows for 95% sentence understanding. To get a better picture of users’ performance, improvement of WRS as well as wearing time should be considered. This will provide us with information regarding improvement in hearing condition with applied treatment, in addition to details about the regions. 

The post-surgery counseling procedure at 3 months might include a discussion about the results achieved based on the HEARRING graph, creating trust and reassurance. 

The HEARRING graph will be further developed from a representation to an active evaluation tool, enabling the quality control and correlation assessment.

## 5. Conclusions

The HEARRING graph provides a visual aid for the interpretation of the efficacy of the treatment. Both elements, the wearing time and the efficacy in terms of speech comprehension in quiet and in noise, are considered together and may allow for a better understanding about the overall performance and the impact of the hearing device on a daily basis. Additionally, the HEARRING graph enables a better correlation and consistency evaluation between different findings. By offering essential information visually, the HEARRING graph might represent a useful medium to identify low performance hearing aid users who may take advantage of alternative solutions, such as MEI, BCI or a cochlear implant.

The cohort of patients shown in the graph is incomplete as it does not include poor or inconsistent outcomes. It is clear that the HEARRING graph has the potential to be expanded to a more comprehensive and detailed presentation tool by using more clinical and subjective findings. In this first publication, however, we aimed to show the simplicity and efficiency of this tool for daily work with patients’ data. 

## Figures and Tables

**Figure 1 jpm-13-00462-f001:**
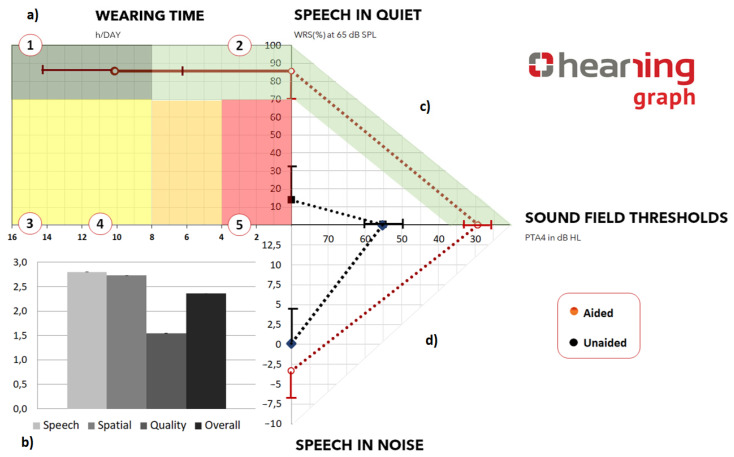
Representative outcomes of 7 subjects [9] (**a**) Top left shows the wearing time (hours/day) as obtained from the patient in relation to WRS. Region 1 indicated that the users achieved ≥70% word recognition score in quiet and wore the device for ≥8 h/day. This exhibited an ideal situation. The patient reached high listening and acceptable wearing comfort. Region 2 represented word recognition score of ≥70% and wearing time of ≤8 h/day. It suggested that an optimal speech comprehension in speech test was achieved but the ideal wearing time was not achieved. Region 3 identified ≤70% word recognition score and wearing time of ≥8 h/day. This indicated ideal wearing time but a need to achieve better speech scores. Region 4 showed ≤70% word recognition score and wearing time between 4–8 h/day. Region 5 marked ≤70% word recognition score and wearing time between 1 and 4 h/day. (**b**) Bottom Left shows the scores (difference between preop and 3 months postop) of SSQ12 questionnaire. (**c**) Top right displays WRS in % at 65 dB SPL in the aided (red dotted lines) as well as unaided (black dotted lines) condition in relation to the respective PTA values on the x axis. (**d**) Bottom right displays speech performance in noise (in dB SNR) in the aided (red dotted lines) as well as unaided (black dotted lines) conditions in relation to the respective PTA values on the x axis.

## Data Availability

The data were extracted from the following paper: [9].

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
