# Peer review of "A Novel Representation of Audiological and Subjective Findings for Acoustical, Bone Conduction and Direct Drive Hearing Solutions"

_jpm, 2023, doi:10.3390/jpm13030462_

Round 1

Reviewer 1 Report

The authors wrote an article about A Novel Representation of Audiological and Subjective Findings for Acoustical, Bone Conduction and Direct Drive Hearing Solutions, introducing HEARRING Graph. The article is quite interesting, the topic is new and this solution can help audiologist to easily understand the hearing aid performance. I have some questions to ask to you.

1. I think that the Satisfaction with Amplification in Daily Life  (SADL) questionnaire and the Abbreviated Profile of Hearing Aid Benefit (APHAB), should give a better subjective contribution, because the personal image with aid it's an important theme to discuss. The HEARRING graph does not help us in this way.

2. I think that you should give a score for each section, and create a final graph to analyze the result, to give an immediate score for the patient satisfaction. 

3. You can analyze in the discussion this article that can give you a better idea how analyze patients satisfaction: Gazia F, Portelli D, Lo Vano M, Ciodaro F, Galletti B, Bruno R, Freni F, Alberti G, Galletti F. Extended wear hearing aids: a comparative, pilot study. Eur Arch Otorhinolaryngol. 2022 Nov;279(11):5415-5422. 

Author Response

Dear Reviewer,

thank you for the constructive input. Here our responses:

  1. I think that the Satisfaction with Amplification in Daily Life  (SADL) questionnaire and the Abbreviated Profile of Hearing Aid Benefit (APHAB), should give a better subjective contribution, because the personal image with aid it's an important theme to discuss. The HEARRING graph does not help us in this way.

ad 1) the available data used to create the HEARRING graph did not include the APHAB or SADL. Only the SSQ was offered to users. The HEARRING graph would allow for any subjective questionnaire to be used. We agree that the APHAB would be particularly interesting also because it's integrated in NOAH. Therefore, the graph could be automatically created by a NOAH module.  

2. I think that you should give a score for each section, and create a final graph to analyze the result, to give an immediate score for the patient satisfaction. 

We have already created a score for each section as well as a final score summarizing the improvements in hearing with treatment. The rational behind the score is based on a formula used already in health care. The concept you mention will be published in the second paper. The second paper will also provide the rational for a quick visual quality control of the audiological findings, thanks to consistency checks among different findings.   

3. You can analyze in the discussion this article that can give you a better idea how analyze patients satisfaction: Gazia F, Portelli D, Lo Vano M, Ciodaro F, Galletti B, Bruno R, Freni F, Alberti G, Galletti F. Extended wear hearing aids: a comparative, pilot study. Eur Arch Otorhinolaryngol. 2022 Nov;279(11):5415-5422. 

Reviewer 2 Report

Summary

The aim of the investigation was to report patient´s audiological as well as subjective outcomes in a single graph, with the overall goal of producing a standardized method for reporting a combination of audiological and subjective outcome measures in clinical practice.

General comments

The idea here of presenting audiological and subjective measures together to get an overview of the overall benefit to the individual of using a hearing aid device is good and interesting. Unfortunately, there are no experimental comparisons made, or attempt to investigate whether the presentation of the measures as reported in the paper results in measurable improvements in assessment. The text of the paper makes it rather confusing regarding why the measures presented were chosen over others that might be available, and the flow of ideas is confusing throughout. For example, line 79 states that “the following findings have been integrated in the graph”, and lists a series of measures. But why these measures? The choice comes across as arbitrary and random – no effort is made to attempt to analyze whether the choice of measures are too many or too few for effective evaluation of the patient, or whether other measures might be better. Why choose to report the SSQ12 questionnaire scores rather than one of the other questionnaires listed in the paragraph starting on line 48? As presented, the text seems to propose that the measures are reported simply because these were the measures reported in study [9], but more justification is needed.

            There does not appear to be any attempt to study various configurations of presentation of the measures in order to establish which is the clearest or most useful. The conclusion on line 207 that “The HEARRING graph provides a visual help for the interpretation of the efficacy of the treatment” appears to be the authors' own subjective viewpoint. No data, evidence, or findings based on statistical tests is provided to support this statement. I am unsure what justification there is in the paper that such a plot offers substantial benefits over looking at individual plots and comparing the data across them. It seems that to do this, a study would need to be designed that offered several different configurations of presentation or choice of measurements, which might then be rated by professionals who work in this area.

            I also have some issues with the HEARRING Graph itself. Figure 1 shows that the SSQ categories (speech, spatial, qualities, and overall) are too small to make out properly, and the y axis is not labelled so the reader does not know what the scale is. There also appears to be wasted space in the subjective plot, where the right half is empty. A substantial amount of empty white space could also potentially be saved on the overall right hand side of the plot – the hearring graph logo, sound field thresholds label and key could probably be moved.

            I do feel that the authors have a good idea here. However, in order to provide an effective means of presenting audiological and subjective measures together to get an overview of the overall benefit to the individual of using a hearing aid device, leading to a standardized method of doing this (as stated in the abstract), a more systematic approach supported by evidence and data analysis would need to be taken, in order to justify the choices made in the presentation of the patient’s measures. This is lacking in the current version of the manuscript.

Specific comments

Line 25: “As an example, the HEARRING graph in this paper…”

            This is the first mention of the HEARRING graph and it is unclear at this stage what this is (is it the same graph as mentioned above on line 21?). Please briefly introduce it to the reader.

Introduction, first paragraph: The text here is rather abrupt, has no citations, and could do with some more detail to help the reader understand what the issues are that the study will address. The first sentence “Audiological measures used in clinical practise may not reflect patient’s overall hearing device benefit” could benefit from a supporting citation. The sentence “Multiple graphical representations are more difficult to read and do not provide a comprehensive interpretation of the outcomes” needs some context – will the current study attempt this? Have any previous studies raised this or attempted to address this? The following text, “Although the HEARRING Graph is not intended to replace any existing means of representation of clinical data, it is aimed to provide an additional comprehensive overview of multiple test findings” also needs more context as the reader is unclear with what the HEARRING GRAPH is just yet (there should be some description), and it is not unclear whether such a graph is able to offer the “comprehensive interpretation of the outcomes” that is mentioned in the previous sentence.

Line 45: “Maier et al introduced minimal reporting standards for middle ear hearing implants in 2018 suggesting standards using the most relevant audiological outcomes [1]. These standards have been used to present the audiological performance in the graph.”

            Please be specific. What graph is being referred to? What standards are being referred to?

Line 48: “To assess subjective benefit, several questionnaires are available, such as AQoL (Assessment of Quality of Life), SSQ (Speech and Spatial Qualities of Hearing), APHAB (Abbreviated Profile Of Hearing Aid Benefit), HUI (Health Utilities Index), DOSO (Device Oriented Subjective Outcome Scale), COSI (Client Oriented Scale of Improvement), HAUQ (Hearing Aid User’s Questionnaire), etc. The aim of these assessment questionnaires is to document patient reported outcomes, focusing on required rehabilitation and overall quality of life.”

            These questionnaires should all have supporting citations.

Line 55: “The Speech, Spatial, and Qualities of Hearing Scale (SSQ) is widely used patient reported questionnaire assessing an individual’s hearing abilities in 3 domains: 1) speech understanding, 2) spatial domain and 3) the hearing quality [2,3].

Among various studies performed recently, wearing time is one factor that has been used very often to measure subjective benefit.”

The text here needs work. It is good that the SSQ has some supporting citations, but why have the authors mentioned this specific questionnaire here? How is the SSQ relevant to the current study? Why go into (limited) detail about it here, while the other questionnaires mentioned above do not get any such detail? How does the sentence about wearing time link to the SSQ paragraph, and what is its relevance? More detail is needed here for the reader to understand what the main points being made here are, as the text in its present form comes across as confusing.

Figure 1 caption. “Representative outcomes of 7 subjects [9].” This is confusing, as the plot shows outcomes for a single subject only. Why not show plots for all 7 subjects, so that the reader can see variation across the plots for different subjects? Ideally, a range of outcomes, both positive and negative, would be shown, to better illustrate how the hearring graph can be used to better assess individual patients with different outcomes.

Line 175: “As hypothesized, the HEARRING graph was designed to represent results from different audiological and subjective tests in one graph.”

            This is incorrect. There was no hypothesis made, and no experiment designed to test any hypotheses is reported.

Line 179: “No new studies were done to generate data for the present study.”

            This is presented as if it were a strength of the study, but it is a substantial weakness. As mentioned above, in order to assess whether the current attempts to represent results from different audiological and subjective tests in one graph were successful or made any significant difference to clinics’ patient data analyses, a study specifically designed to address this question would need to be reported, with data and evidence presented to support the authors conclusions.

Discussion section: This section does not include a single reference to previous work, so the current results cannot be put in the context of existing work. The section is full of statements that are vague and unsupported by evidence e.g. “The graph may help in assessing reliability of tests, understanding patient needs, setting realistic goals for patients, changing or modifying treatment, counseling patients, planning effective rehabilitation and motivating patients.” Why? How is the graph likely to help in assessing reliability of tests? What is the evidence to support this statement? It seems just as likely that existing graphs might be as effective, and the introduction of the current plots in a clinical setting might make no difference to the outcomes for patients.

Line 205: “The HEARRING graph will be further developed from a representation to an active evaluation tool enabling the quality control and correlation assessment.”

            This is too vague. What is meant by “active?” What measures are meant to be correlated and why?

Author Response

Dear Reviewer,

thank you for your thorough and critical review of our manuscript. The aim of this paper is to propose a novel representation graph displaying both, subjective and objective outcomes. The authors do not intend to allow for a statistical analysis with this graph. The intention was to make it easier for clinical personnel (ENT doctor, Audiologist or even Implant Technician) to read multiple outcomes to provide the patient on the spot with a comprehensive interpretation of his/her outcomes. It is also intended to offer a quick quality control of the data on a single user level (ie. If there is a possible masking issue etc.). Our aim was to display the importance of daily wearing time and its correlation with the audiological benefit (proven fact), hence the subjective satisfaction of the patients. The authors decided to only present one data set of one of the co-authors recently published outcomes to focus on the novelty of the data presentation and not on the novelty of the results.

Any QoL questionnaire can be introduced into the graph – therefore some possibilities of QoL questionnaires are mentioned in the manuscript.

The graph regarding the SSQ12 questionnaire was adapted as requested – please see new graph.

Best regards

Reviewer 4 Report

1. Title - Clearly defines the work.

They present a new representation of the audiological and subjective benefits felt by the patient

2 - The proposal for studying this topic is very interesting.

3 - Abstract - It is well elaborated, succinct and summarizes the article.

4 - Introduction - It is appropriate to the topic

5 - Materials and methods - Clearly explains the methodology used,

6 Discussion - The discussion and conclusions highlight the possible benefits of using this representation of results

7 - References are adapted and updated.

8 - The tables and figures are well organized and didactic for a better understanding of the text.

Round 2

Reviewer 2 Report

I still feel that the idea of presenting audiological and subjective measures together to get an overview of the overall benefit to the individual of using a hearing aid device is good and interesting. Unfortunately, the authors’ response is inadequate, and does not sufficiently address the comments made in the previous review. Looking at the latest version of the submitted paper, the main changes made in response to the reviewers in the main text have been to add a sentence and a citation. This is clearly not sufficient. The authors’ response letter is too short and does not properly respond to the points made, even the relatively simple and straightforward ones e.g. For the point [“Line 25: “As an example, the HEARRING graph in this paper…”            This is the first mention of the HEARRING graph and it is unclear at this stage what this is (is it the same graph as mentioned above on line 21?). Please briefly introduce it to the reader.] No attempt has been made to address this. As a result, all the issues with the non-scientific approach taken by the authors that was raised in the previous round of reviews remain. Specifically: [“The HEARRING graph provides a visual help for the interpretation of the efficacy of the treatment” appears to be the author’s own subjective viewpoint. No data, evidence, or findings based on statistical tests is provided to support this statement.] is still a major problem. The authors appear to be unwilling to address these shortcomings. My recommendation is therefore Reject.